# Peptidomimetic blockade of MYB in acute myeloid leukemia

Kavitha Ramaswamy[1,2], Lauren Forbes[1,3], Gerard Minuesa[1], Tatyana Gindin[4], Fiona Brown[1], Michael G. Kharas [1], Andrei V. Krivtsov[5,6], Scott A. Armstrong[2,5,6], Eric Still[1], Elisa de Stanchina[7], Birgit Knoechel[6], Richard Koche[5] & Alex Kentsis [1,2,3]

Aberrant gene expression is a hallmark of acute leukemias. MYB-driven transcriptional coactivation with CREB-binding protein (CBP)/P300 is required for acute lymphoblastic and myeloid leukemias, including refractory MLL-rearranged leukemias. Using structure-guided molecular design, we developed a peptidomimetic inhibitor MYBMIM that interferes with the assembly of the molecular MYB:CBP/P300 complex and rapidly accumulates in the nuclei of AML cells. Treatment of AML cells with MYBMIM led to the dissociation of the MYB:CBP/P300 complex in cells, its displacement from oncogenic enhancers enriched for MYB binding sites, and downregulation of MYB-dependent gene expression, including of MYC and BCL2 oncogenes. AML cells underwent mitochondrial apoptosis in response to MYBMIM, which was partially rescued by ectopic expression of BCL2. MYBMIM impeded leukemia growth and extended survival of immunodeficient mice engrafted with primary patient-derived MLL-rearranged leukemia cells. These findings elucidate the dependence of human AML on aberrant transcriptional coactivation, and establish a pharmacologic approach for its therapeutic blockade.

[1] Molecular Pharmacology Program, Sloan Kettering Institute, New York, NY 10065, USA. [2] Department of Pediatrics, Memorial Sloan Kettering Cancer Center, New York, 10065 NY, USA. [3] Departments of Pediatrics, Pharmacology, and Physiology & Biophysics, Weill Cornell Medical College, Cornell University, New York, NY 10065, USA. [4] Department of Pathology and Cell Biology, Columbia University Medical Center and New York Presbyterian Hospital, New York, NY 10065, USA. [5] Center for Epigenetics Research, Sloan Kettering Institute, New York, NY 10065, USA. [6] Department of Pediatric Oncology, Dana-Farber Cancer Institute, Boston, MA 02215, USA. [7] Antitumor Assessment Core Facility, Memorial Sloan Kettering Cancer Center, New York, NY 10065, USA. Correspondence and requests for materials should be addressed to A.K. (email: kentsisresearchgroup@gmail.com)

Despite recent efforts to improve stratification of conventional chemotherapy for the treatment of patients with acute myeloid leukemia (AML), survival rates remain less than 70% and 40% for children and adults, respectively[1,2]. Recent genomic profiling studies have begun to reveal that AML is characterized by the predominance of mutations of genes encoding regulators of gene transcription and chromatin structure[3,4]. Indeed, most AML chromosomal translocations, such as those involving *MLL* (*KMT2A*) gene rearrangements, encode chimeric transcription or chromatin remodeling factors[5]. Recent functional genomic efforts have identified specific molecular dependencies of aberrant AML gene expression, such as the requirement of *DOT1L* for the maintenance of *MLL*-rearranged leukemias, prompting the clinical development of DOT1L methyltransferase inhibitors for AML therapy[6,7]. Similarly, additional AML subtypes appear dependent on aberrant regulation of gene expression, conferring a susceptibility to inhibition of CDK8 and BRD4 that in part regulate the Mediator transcriptional coactivation complex[8-10].

In addition, recent studies have also implicated aberrant activity of hematopoietic transcription factors and their co-activators, such as MYB and CBP/P300, in recruitment of the basal transcriptional apparatus in AML cells[8,11,12]. In particular, MYB is a sequence-specific hematopoietic transcription factor that is translocated and aberrantly duplicated in a subset of T-cell acute lymphoblastic leukemias (T-ALL)[13,14]. Leukemogenic activities of MYB require its physical and specific association with the transcriptional co-activator CBP and its nearly identical paralogue P300[11]. This interaction is associated with the recruitment of CBP/P300 and its chromatin remodeling of transcriptional circuits required for leukemogenesis[15].

While CBP/P300 can be inactivated by nonsense and missense mutations in a variety of cancers including acute lymphoblastic leukemias[16], both MYB and CBP/P300 are not currently known to be mutated in AML[17]. Importantly, transient suppression of MYB expression can eliminate *MLL-AF9* leukemias but is dispensable for normal myelopoiesis, emphasizing its specific functional requirements in AML pathogenesis[8]. In addition, the *Booreana* strain of mice that is mutant for Myb E308G in its transcriptional activation domain and impairs the molecular recognition of Myb by the KIX domain of Cbp/p300, exhibits normal hematopoiesis, but is resistant to leukemogenesis induced by the *MLL-AF9* and *AML1-ETO* oncogenes[11]. Altogether, these considerations raise the possibility that blockade of aberrant transcriptional coactivation by CBP/P300 and its transcription factors may constitute a therapeutic strategy in AML.

Previous attempts to interfere with aberrant transcriptional coactivation in AML have focused on the pharmacologic blockade of lysyl acetyltransferase activities of CBP/P300[18,19]. In addition, chetomin and napthol derivatives have been identified to interfere with the protein–protein interactions of the MYB–CBP/P300 complex[20-22]. Here, we extended these efforts by focusing on the specific requirement of MYB E308 in its transcriptional activation domain for molecular recognition of the CBP/P300 KIX domain to therapeutically target and dismantle the assembly of the MYB:CBP/P300 leukemogenic transcription factor–coactivator complex, as hypothesized previously[11,23,24]. Using molecular dynamics simulations and structural analysis of the MYB:CBP/P300 molecular complex, we designed a stabilized, cell-penetrant peptidomimetic inhibitor of MYB:CBP/P300 binding, termed MYBMIM. Consequently, we investigated its molecular and cellular activities, blockade of leukemogenic gene expression, and therapeutic potential in preclinical leukemia models in vitro and in vivo.

## Results

### Design and binding activity of peptidomimetic MYB:CBP inhibitor MYBMIM.

Stereoselective substitution of D-amino acids in peptides and their fusion to protein transduction domains have been used to enhance their stability and intracellular delivery, respectively[25,26]. On the basis of the importance of the Myb E308 residue for MYB:CBP/P300 binding and leukemic transformation[11,23,24], we reasoned that a peptide designed to compete with this region of MYB might represent an effective therapeutic inhibitor. We thus developed a peptide mimetic of MYB residues 293–310, based on the high-resolution structure of the MYB:CBP/P300 complex (Fig. 1a). We fused this peptide to the cationic cell-penetrant TAT peptide, as optimized by Dowdy and colleagues[27-30]. The peptide was designed in the retro-inverso orientation containing D-amino acids, and termed MYBMIM (Fig. 1b, Supplementary Table 1). As retro-inverso strategies are able to mimic selected helical peptides[31,32], we used molecular dynamics simulations to model the binding of the retro-inverso and native forms of MYB peptides to the CBP/P300 KIX domain (Fig. 1b). This analysis revealed that the retro-inversion of MYB peptide stereochemistry is compatible with binding to the CBP/P300 KIX domain, as evidenced by the largely complete preservation of key MYB:CBP/P300 contacts, including the E308:H602 and R294:E665 salt bridges, and the L302 hydrophobic burial (Supplementary Fig. 1). We also designed inactive versions of MYBMIM, termed TG1, TG2, and TG3 (Supplementary Table 1), that are identical to MYBMIM with the exception of substitutions of R294G, L302G, and/or E308G residues that make key contacts with CBP/P300, as identified from molecular dynamics simulations (Fig. 1a and b, Supplementary Fig. 1). Using microscale thermophoresis, we empirically measured binding affinities of MYBMIM, its L-amino acid containing counterpart MYB, TG1, TG2, and TG3 to the purified recombinant CBP KIX domain, as compared to the control TAT peptide (Fig. 1c). We observed that MYBMIM bound to the CBP KIX domain in a MYB, not TAT, peptide-dependent manner, albeit with a slightly reduced binding affinity as compared to the L-amino acid peptide, consistent with the expected effects of retro-inversion. The TG1, TG2, and TG3 analogs exhibited progressively reduced affinities to the CBP KIX domain, consistent with the destabilizing effects of their substitutions (Fig. 1c). TG3 showed the lowest affinity to the CBP KIX domain, confirming that it is suitable as an inactive analog of MYBMIM. Using live cell confocal fluorescence microscopy of fluorescein isothiocyanate (FITC)-conjugated MYBMIM peptide, we confirmed rapid MYBMIM accumulation in the nuclei of MLL-rearranged MV-411 AML cells (Fig. 1d). These results suggest that MYBMIM may constitute an approach for the pharmacologic blockade of MYB:CBP/P300 transcriptional coactivator complex in leukemia cells.

To test this hypothesis directly, we immobilized biotinylated forms of MYBMIM (BIO-MYBMIM) on streptavidin-conjugated beads (Supplementary Table 1), and used them to affinity-purify CBP/P300 from native cellular extracts of MLL-rearranged MV-411 cells (Fig. 1e). Consistent with the computational and empiric binding studies (Fig. 1b and c), we observed efficient and specific binding of BIO-MYBMIM to CBP/P300 in cellular extracts, as evidenced by the displacement of cellular CBP/P300 by competition with excess of free MYBMIM, but not by the retro-inverso TAT control peptide (RI-TAT, Fig. 1e). To determine the ability of MYBMIM to dissociate the MYB:CBP/P300 complex in AML cells, we purified the MYB:CBP/P300 complex by immunoprecipitation using specific anti-MYB antibodies in the presence of 0 or 20 μM free MYBMIM, and determined its composition by western immunoblotting

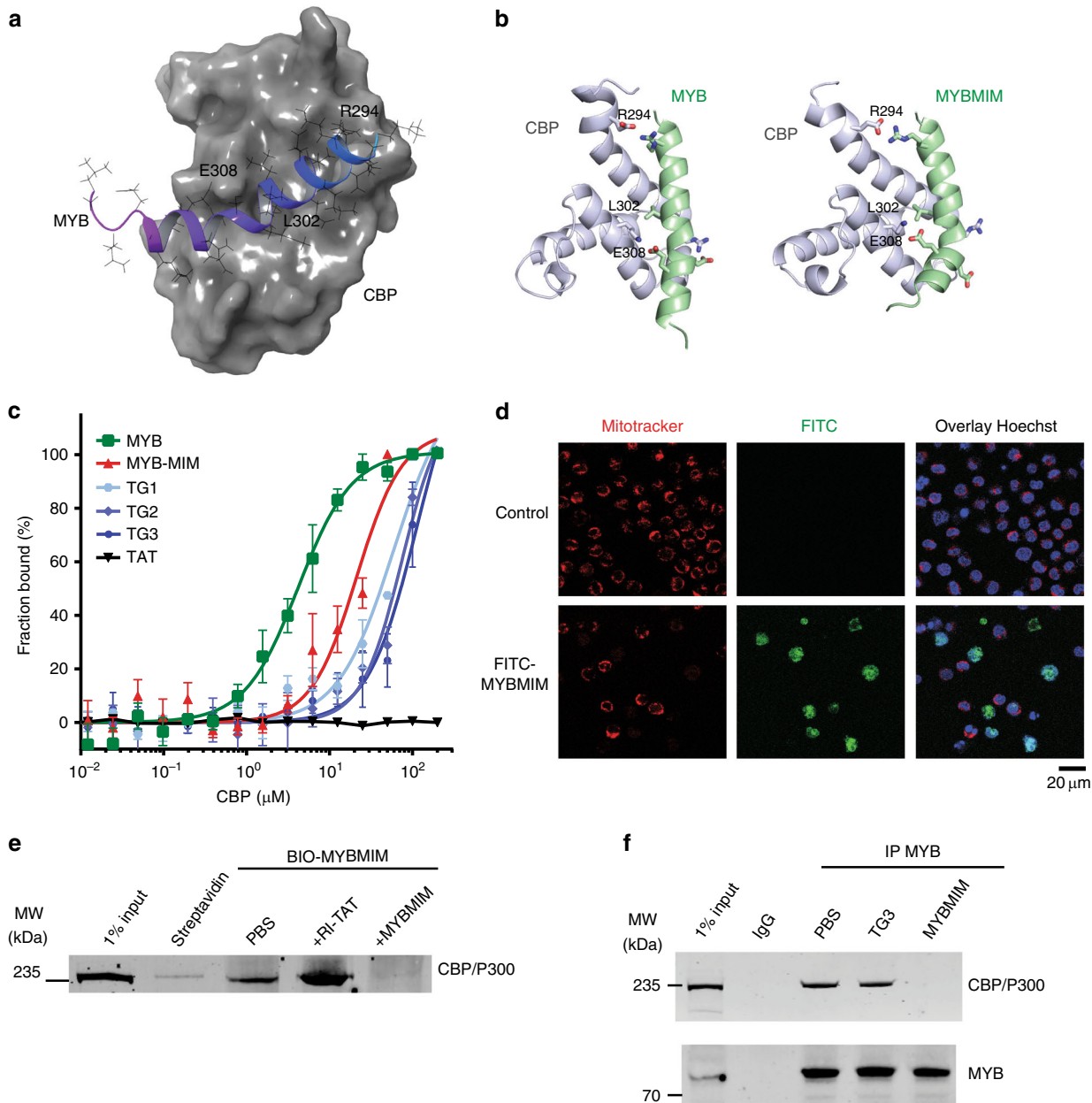

**Fig. 1** MYBMIM disrupts the MYB:CBP complex in AML cells. **a** Molecular structure of the complex of the transactivation domain of MYB (blue) with the KIX domain of CBP (gray)[23] assembled in Maestro (Schrödinger). MYB residues making contacts with CBP are labeled as indicated. (PDB: 1SB0). **b** Molecular structures of the transactivation domain of MYB (left, green) and MYBMIM (right) in complex the KIX domain of CBP (gray), as modeled using replica exchange molecular dynamics. Both MYBMIM and MYB retain E308 and R294 salt bridge and L302 hydrophobic interactions, as marked by sidechain representation. **c** Binding of FITC-conjugated MYB (green), MYBMIM (red), compared to control TG1, TG2, TG3, and TAT (black), as measured using microscale thermophoresis ($K_d = 4.2 \pm 0.5\,\mu M$ and $21.3 \pm 2.9\,\mu M$ for MYB and MYBMIM, respectively, $59.2 \pm 12.4\,\mu M$ for TG1, $75.1 \pm 12.5\,\mu M$ for TG2 and $113.5 \pm 36.6\,\mu M$ for TG3). Error bars represent standard error mean of three biological replicates. **d** Live cell confocal fluorescence microscopy photographs of MV-411 cells treated with 50 nM FITC-MYBMIM (green) for 1 h, as visualized using Mitotracker (red) and Hoechst 33342 (blue). Scale bar indicates 20 μm, with z-stack of 1.5 μm. **e** Western blot showing comparable binding of cellular CBP/P300 to streptavidin bead-immobilized BIO-MYBMIM, specifically competed by 20-fold excess free retro-inverso TAT (RI-TAT) and MYBMIM peptides, as indicated by + signs. **f** Representative western blot of MYB:CBP/P300 complex immunoprecipitated from MV411 cells disrupted 20 μM MYBMIM and TG3, as indicated

(Fig. 1f). We found that MYBMIM competition led to significant dissociation of the cellular MYB:CBP/P300 complex, as compared to untreated or control treated complexes (Fig. 1f), consistent with the competitive binding affinities of the retro-inverso MYBMIM and native MYB peptides to the CBP KIX domain in vitro (Fig. 1c). Thus, MYBMIM is a specific peptidomimetic inhibitor of MYB:CBP/P300 complex assembly in cells.

**MYBMIM suppresses transcriptional enhancers and activation in AML cells.** MYB and CBP/P300 mediate their transcriptional co-activation effects in part through the assembly and stabilization of transcription factor complexes at specific enhancers and promoter elements[33,34]. Thus, dissociation of the MYB:CBP/P300 complex by MYBMIM would be expected to reduce MYB-dependent occupancy and gene trans-activation at specific target genes responsible for aberrant leukemia cell growth and

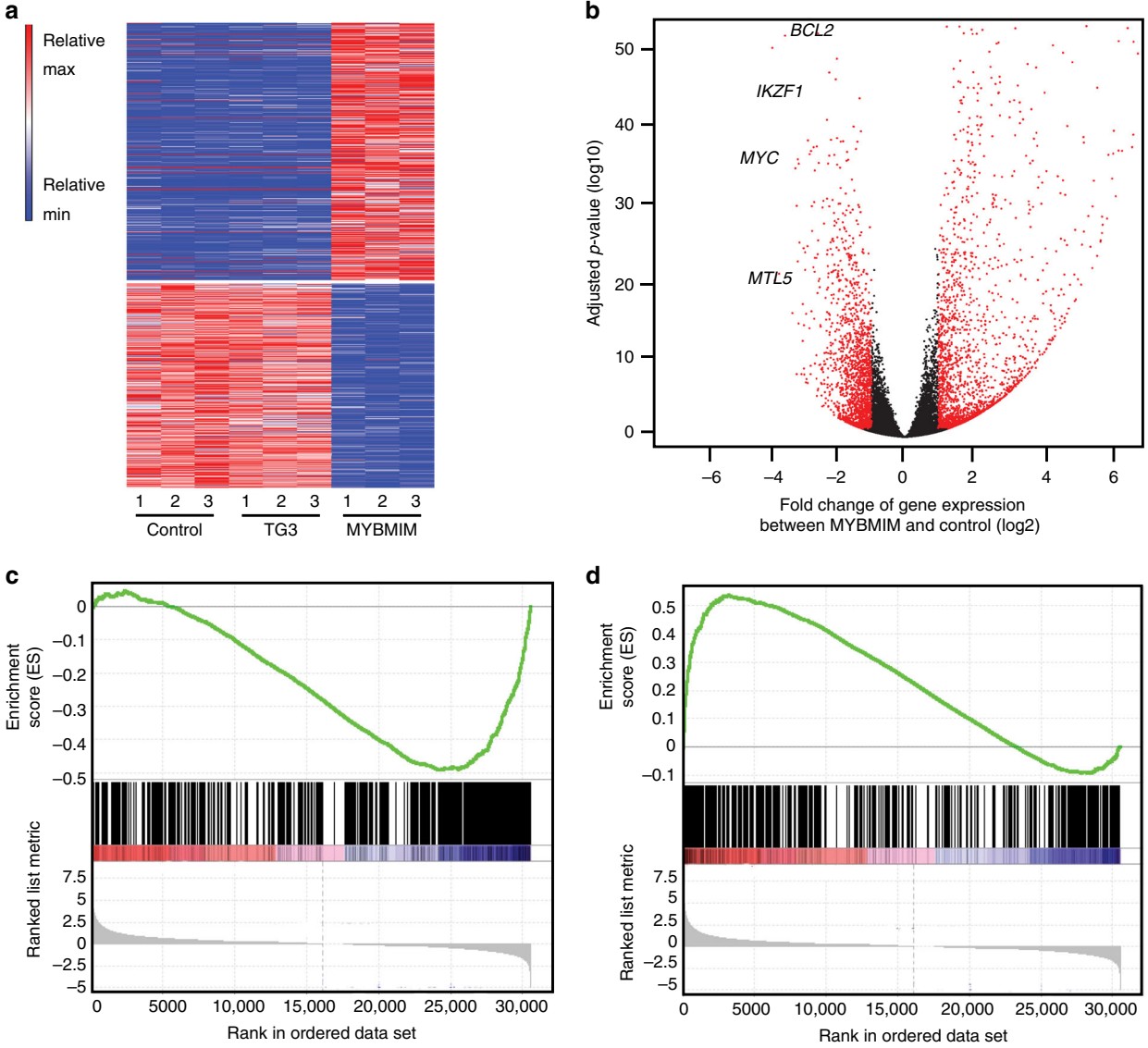

**Fig. 2** MYBMIM regulates MYB enhancers and promoters and MYB-dependent target genes. **a** Heatmap of changes in normalized gene expression of MOLM13 cells treated with 20 μM MYBMIM versus TG3 control for 6 h, as analyzed by RNA-seq of three biological replicates. **b** Volcano plot of normalized gene expression, with *BCL2, IKZF1, MYC and MTL5* as indicated. **c**, **d** Gene set enrichment analysis of downregulated **c** and upregulated **d** genes with respect to MYB target genes, as defined by[8]. NES = −2.47 and 2.09, and $q = 0$ and 0, respectively

survival. To investigate the effects of MYBMIM on gene expression in AML cells, we analyzed transcriptome profiles of MLL-rearranged MOLM-13 cells treated with MYBMIM as compared to TG3 control using RNA sequencing (RNA-seq). We observed no significant changes in gene expression induced by TG3 as compared to mock-treated cells, confirming the specificity of MYBMIM-induced effects (Fig. 2a). In contrast, we observed that treatment with MYBMIM induced significant downregulation of *BCL2, MYC, GFI1, MTL5, IKZF1* gene expression (Fig. 2b, Supplementary Data 1), in agreement with prior studies of MYB-regulated genes in myeloid cells[35]. In addition to a total of 1730 significantly downregulated genes, we also observed a total of 2232 genes that were significantly upregulated upon MYBMIM treatment, consistent with previous reports of MYB-induced gene repression[35]. Notably, the genes affected by MYBMIM treatment exhibited significant enrichment for direct MYB target genes, as defined by prior studies[8] (Fig. 2c and d). Thus, MYBMIM blocks MYB-dependent leukemogenic gene expression in AML cells.

To test the prediction that MYBMIM would suppress the assembly of MYB:CBP co-activation complexes, we used specific chromatin factor immunoprecipitation followed by DNA sequencing (ChIP-seq) to analyze genome-wide distribution of MYB protein complexes in MV-411 cells treated with MYBMIM. We found that treatment with MYBMIM, but not with its near-isosteric inactive TG3 analog or untreated control, led to the elimination of 2,690 MYB complexes bound to promoters and enhancers (Fig. 3a, Supplementary Data 2). Of the total 5,122 MYB protein complex-bound loci, 587 were found to occur within 50 kb of the 1730 significantly downregulated genes observed in coupled transcriptome analyses (Supplementary Fig. 2). In addition, we found that MYB-bound promoters and enhancers, specifically affected by MYBMIM treatment as compared to TG3 or untreated controls, were significantly enriched for DNA sequence motifs corresponding to MYB, ERG, SPI1/PU.1, CEBPA, and RUNX1 transcription factors (Supplementary Fig. 3, Supplementary Table 2). This suggests

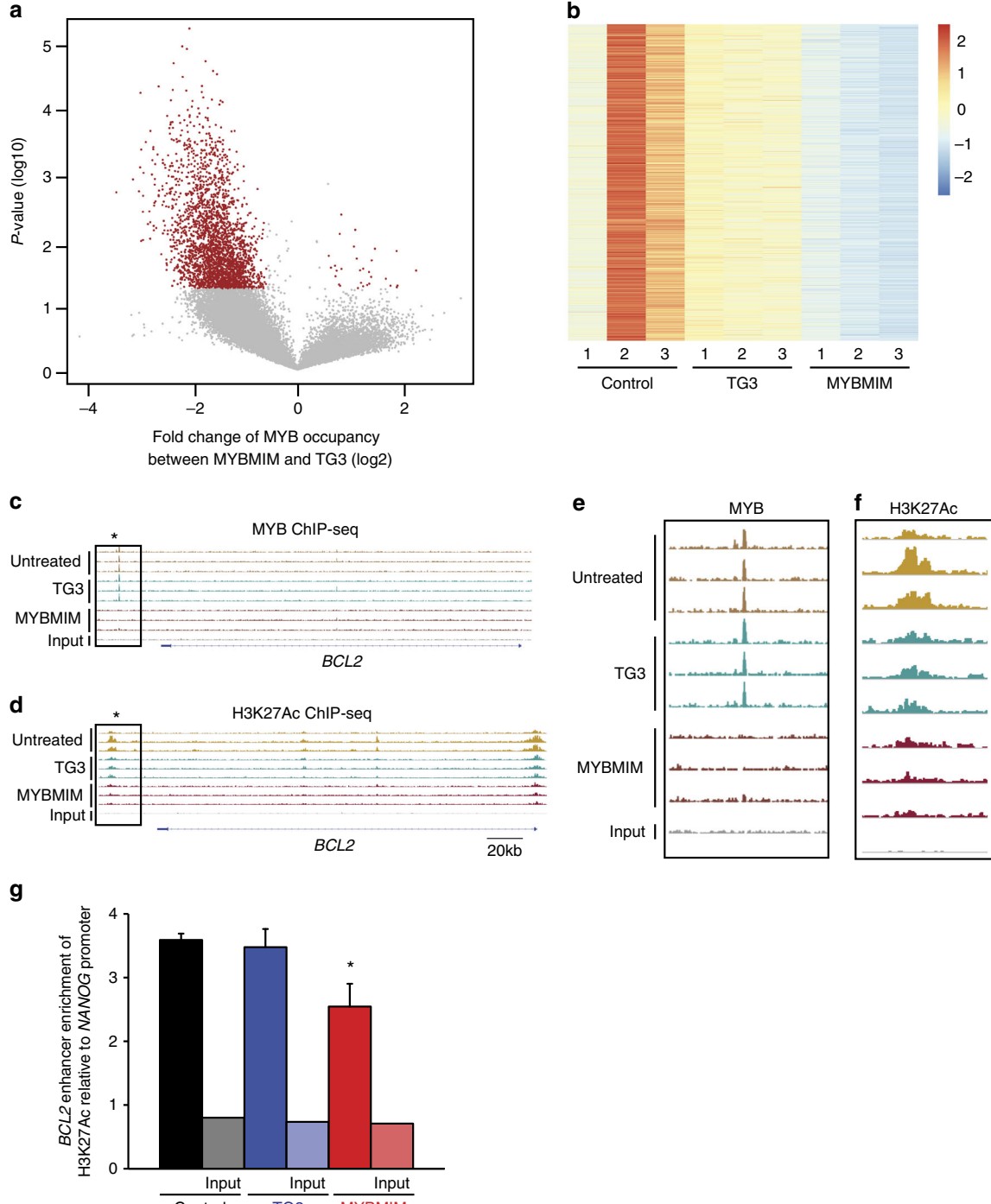

**Fig. 3** MYBMIM suppresses the assembly of chromatin complexes. **a** Volcano plot of MYB occupancy in MV-411 cells treated with 20 μM MYBMIM versus TG3 control for 6 h, as analyzed by MYB ChIP-seq. p-values denote t-test statistical significance of 3 biological replicates. **b** Heatmap of changes in H3K27Ac occupancy of MV411 treated with 20 μM MYBMIM versus TG3 control for 24 h, as analyzed by ChIP-seq of three biological replicates. **c** Genome track of the *BCL2* locus showing elimination of the MYB-bound enhancer (star) upon treatment with MYBMIM, but not control or TG3 treatment. **d** Genome track of the *BCL2* locus showing elimination of the H3K27Ac-bound enhancer (star) upon treatment with MYBMIM, but not control or TG3 treatment. **e** Magnified boxed area of MYB-bound enhancer peak shown in 3c. **f** Magnified boxed area of H3K27Ac-bound enhancer peak shown in (**d**). **g** Analysis of relative enrichment of H3K27Ac at the *BCL2* enhancer locus compared to *NANOG*, as measured by ChIP-PCR upon treatment with control PBS (black), 20 μM TG3 (blue), and 20 μM MYBMIM (red) for 24 h. Error bars represent standard deviations of three biological replicates. * p = 8.6e–3 when compared to untreated control

that their DNA binding may cooperate with MYB and/or CBP/P300, as suggested by prior studies[15].

A key mechanism of CBP/P300 co-activation involves its acetylation of K27 of histone 3 (H3K27Ac), which facilitates gene trans-activation[34,36]. To examine the effects of MYBMIM on

CBP/P300-associated histone acetylation, we analyzed H3K27Ac genome-wide using ChIP-seq methods and noted a significant reduction in MYB-containing H3K27Ac sites by MYBMIM treatment as compared to TG3 control (52% reduction, p = 0.0032, median fold change). This difference was in spite of

the genome-wide reduction in all H3K27Ac sites (33% reduction, $p = 0.034$). There were a total of 1,479 sites with significantly decreased H3K27Ac enrichment (Fig. 3b, Supplementary Data 3). We then focused on the enhancer at the *BCL2* locus, where we observed significant reduction of both MYB binding and H3K27 acetylation by MYBMIM treatment, as compared to untreated cells or cells treated with the inactive TG3 analog (Fig. 3c–f). Using chromatin immunoprecipitation followed by quantitative genomic PCR (ChIP-qPCR), we observed a significant albeit incomplete reduction of H3K27Ac at the MYBMIM-displaced *BCL2* enhancer in cells treated with MYBMIM as compared to TG3 or untreated control ($p = 8.6e-3$, $t$-test, Fig. 3g), as well as other known MYB target genes, such as *GFI1* (Supplementary Fig. 4). Thus, MYBMIM suppresses transcriptional enhancers and activation in AML cells.

**MYBMIM induces apoptosis of AML cells in vitro**. As AML cells require MYB:CBP/P300-dependent gene expression for growth and survival, we reasoned that MYBMIM should exhibit growth suppressive effects on AML cells. Using MYBMIM doses similar to the binding affinities using direct biochemical assays (Fig. 1c), we treated a panel of AML cell lines with MYBMIM, including those with (MOLM-13 and MV-411) and without *MLL* rearrangements (ML-2 and HL-60). We observed that MYBMIM, but not its inactive congeners TG1, TG2, or TG3, induced sustained, logarithmic reduction of growth of AML cell lines when compared to untreated control ($p = 7.2e-7$ for MOLM-13; $p = 9e-6$ for MV-411, $p = 1.7e-6$ for ML-2, $p = 3.3e-6$ for HL-60, $t$-test, Fig. 4a). No significant differences in cell growth or viability were observed upon treatment with L-amino acid containing peptides, consistent with their expected proteolysis in cells and media[37] (Supplementary Fig. 5). We did not observe significant changes in the morphologic differentiation of MYBMIM-treated cells (Fig. 4b), with no significant changes in monocytic CD14, granulocytic CD66b, and monocytic CD11b expression, as measured by flow cytometry (Supplementary Fig. 6).

On the other hand, MYBMIM treatment induced significant apoptosis, as assessed by cell surface annexin V and intracellular caspase 3 cleavage by flow cytometry ($p = 5.4e-3$, $t$-test, Fig. 4c, Supplementary Fig. 7). Since MYBMIM treatment induced apoptosis and downregulated *BCL2*, we reasoned that downregulation of *BCL2* expression may be in part but not entirely responsible for the apoptotic effects of MYBMIM on AML cells. We used quantitative reverse transcriptase-polymerase chain reaction (qRT-PCR) to confirm that *BCL2* expression was significantly downregulated by more than two-fold by MYBMIM, but not TG3 or mock treatment in MV-411 and MOLM-13 cells ($p = 4.6e-3$ and $p = 8.2e-4$ for *BCL2* and *MYC*, respectively, Fig. 4d, Supplementary Fig. 8) and confirmed a significant albeit modest decrease in protein abundance of BCL2 upon MYBMIM treatment (Supplementary Fig. 9). To determine if *BCL2* downregulation is necessary for MYBMIM-induced apoptosis of AML cells, we expressed *BCL2* using MSCV-IRES-GFP (MIG) retrovirus in MV-411 cells, and confirmed its ectopic overexpression using qRT-PCR (Supplementary Fig. 10). Consistently, MYBMIM, but not its inactive TG3 analog, induced significant reduction of cell growth and survival of mock-treated and MIG empty vector control transduced MV-411 cells ($p = 0.003$, Fig. 4e). In contrast, cells ectopically overexpressing *BCL2* were largely, though not completely, rescued from MYBMIM-induced apoptosis (Fig. 4e). Although we cannot exclude the possibility of as of yet unknown cellular factors or components displaced by MYBMIM from the MYB:CBP/P300 complex, the disassembly of this complex does appear to contribute to MYBMIM-induced apoptosis. Thus, MYBMIM impairs AML cells growth and

survival in vitro, at least in part by downregulating anti-apoptotic *BCL2* gene expression.

**MYBMIM impedes human leukemia progression in mouse xenograft models in vivo**. To investigate the potential of MYBMIM for leukemia therapy, first we analyzed the effects of MYBMIM on the proliferation and differentiation of healthy human umbilical cord blood (HUCB) hematopoietic progenitor cells in vitro and steady-state mouse hematopoiesis in vivo. We isolated CD34+ mononuclear HUCB cells, and assessed their self-renewal and multi-lineage differentiation using clonogenic assays in methylcellulose in vitro[38]. We observed no significant effects on granulocyte/macrophage and erythroid progenitors, as assessed by their morphology and clonogenicity (Fig. 5a and b). Likewise, we observed no significant changes in peripheral blood counts of C57BL/6J mice treated with MYBMIM by daily intraperitoneal (IP) injection for 7 days, as measured by the analysis of total leukocytes, lymphocytes, platelets, and blood hemoglobin (Fig. 5c–f). Thus, transient MYBMIM exposure is compatible with normal hematopoiesis. To assess the pharmacokinetics of MYBMIM, C57BL/6J mice were treated with a single dose of 25 mg/kg BIO-MYBMIM by IP injection, and their plasma was analyzed for BIO-MYBMIM at varying time points post-injection. The concentration of BIO-MYBMIM was measured by spectrophotometric avidin assay, and results showed biphasic elimination, with peak plasma peptide levels being reached by 30 min post injection followed by a second slow elimination phase (Fig. 5g). These results led us to a dosing regimen of 25 mg/kg twice daily for in vivo studies.

To investigate the anti-leukemia efficacy of MYBMIM, we engrafted sublethally irradiated NOD-scid IL2Rγnull (NSG) mice with primary patient-derived MLL-rearranged human leukemia cells, with their detailed characterization described in Supplementary Table 3, and determined leukemia development using peripheral blood flow cytometry for human-specific CD45 (hCD45). Moribund mice were sacrificed, and human leukemia cells were transplanted for propagation and therapeutic studies using two different treatment paradigms: (i) mice with high burden of disease and circulating leukemia cells, and (ii) mice with residual disease. First, upon leukemia development in tertiary recipients, as defined by at least 1% hCD45-positive cells circulating in peripheral blood, mice were randomized to receive intraperitoneal MYBMIM (25 mg/kg twice daily) or vehicle control daily for 21 days. At the completion of treatment, MYBMIM-treated mice exhibited a significant reduction in leukemia burden, as assessed by bone marrow analysis of human leukemia cells ($p = 1.2e-4$, log-transformed $t$-test, Fig. 5h, i). We assessed levels of BCL2 in the residual leukemia cells in the bone marrow using quantitative immunofluorescence, and found that MYBMIM-treated mice exhibited minimal reduction of levels of BCL2 as compared to vehicle treated mice, without reaching statistical significance ($p = 0.3$, log-transformed $t$-test, Fig. 5j, k). In an independent experiment, we transplanted NSG mice with primary MLL-rearranged human leukemia cells, and treated engrafted animals 3 days post-transplantation with intraperitoneal MYBMIM (25 mg/kg twice daily) or vehicle control for 14 days. Mice were subsequently followed for the development of overt leukemia and survival. We observed that MYBMIM treatment significantly delayed leukemia progression and extended survival ($p = 3.8e-3$, log-rank, Fig. 5l) without causing significant weight loss (Supplementary Table 5). Consistent with the function of MYB in leukemia stem cell maintenance, leukemia cells obtained from moribund mice treated with MYBMIM as compared to vehicle control, exhibited significantly delayed disease latency in secondary transplant recipients ($p < 0.0001$,

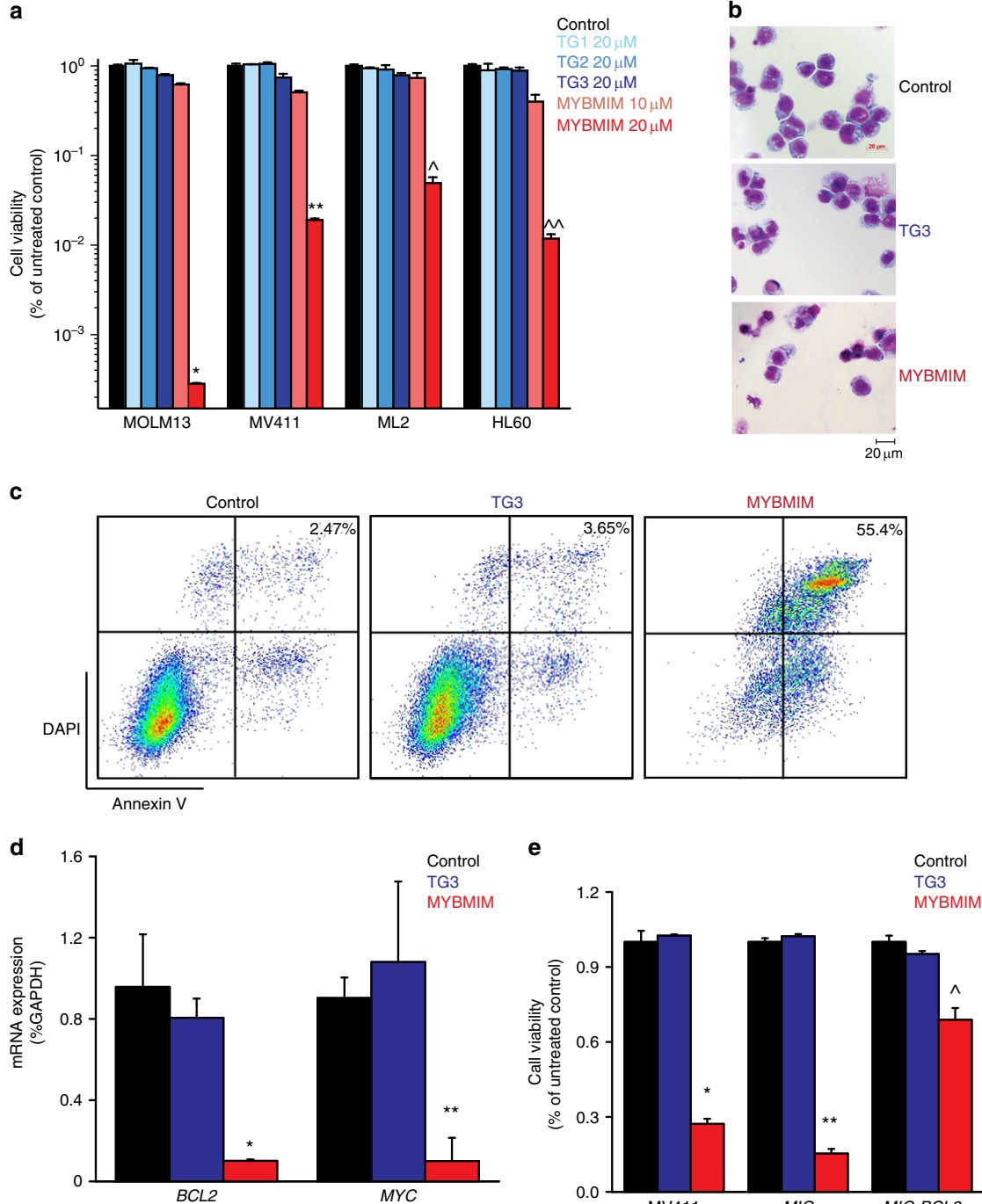

**Fig. 4** MYBMIM induces apoptosis and downregulates MYB-regulated genes. **a** Viability of MOLM-13, MV-411, ML2, and HL60 cells, treated for 6 days with control PBS (black), 20 μM TG1, TG2, or TG3 (blue), and 10 μM MYBMIM (orange) and 20 μM MYBMIM (red), with peptide replacement every 48 h. Error bars represent standard deviations of three biological replicates. *$p = 7.2e{-}7$; **$p = 9e{-}6$; ˆ$p = 1.7e{-}6$; ˆˆ$p = 3.3e{-}6$ when compared to untreated control. **b** Representative photographs of Giemsa-stained MV-411 cells after 6 h treatment, as indicated. Scale bar corresponds to 20 μm. **c** Flow cytometry analysis of apoptosis of MV-411 cells upon peptide treatment at 20 μM for 24 h, as indicated. Numbers denote percentage of cells that are both Annexin V and DAPI positive. **d** Analysis of *BCL2* and *MYC* mRNA expression in MV411 cells as measured by qRT-PCR, upon treatment with control PBS (black), 20 μM TG3 (blue), and 20 μM MYBMIM (red) for 6 h. Error bars represent standard deviations of three biological replicates. *$p = 0.0046$; **$p = 0.008$ when compared to untreated control. **e** MV-411 cells expressing MSCV-IRES-GFP (MIG) BCL2 but not empty MIG or wild-type cells are protected from treatment with 20 μM MYBMIM (red) as compared to control PBS (black) and 20 μM TG3 (blue) peptides. Error bars represent standard deviations of three biological replicates. *$p = 1.4e{-}5$; **$p = 4.2e{-}7$; ˆ$p = 0.0005$ MYBMIM treatment compared to respective untreated controls

log-rank, Fig. 5m). Thus, MYBMIM exhibits therapeutic anti-leukemia efficacy in preclinical AML mouse models in vivo.

## Discussion

Transcriptional co-activation is increasingly recognized as a fundamental process controlling physiologic gene expression in normal cell development and its dysregulation in cancer cells. In particular, acute myeloid leukemias, blood cancers that remain difficult to treat in spite of intensive combination chemotherapy and stem cell transplantation, are often caused by mutations of genes encoding factors that regulate gene expression. Similar mechanisms appear to be dysregulated in a large fraction of

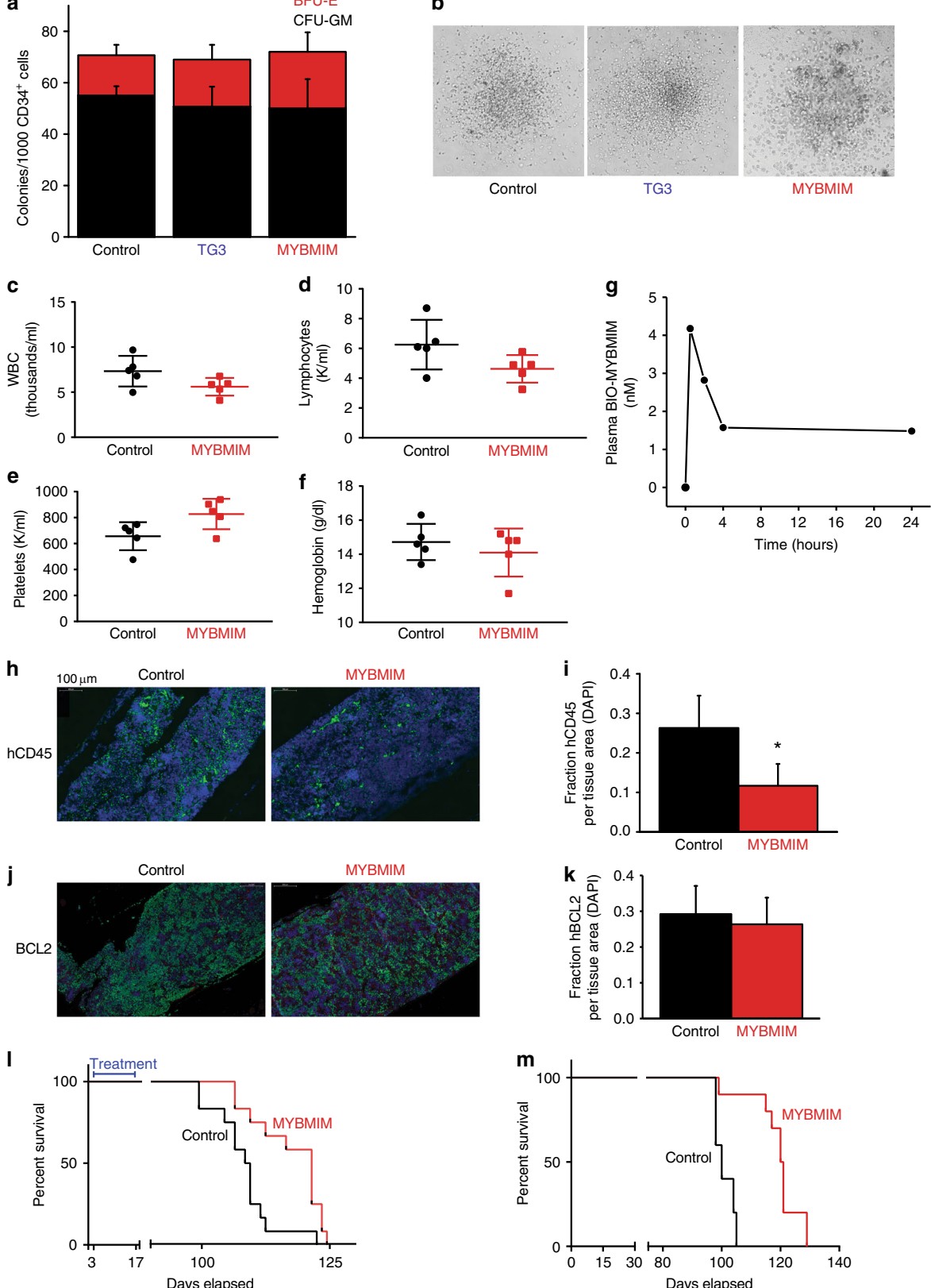

human cancers, at least in part due to the convergence of developmental and oncogenic gene expression in cell fate specification and development[39]. As such, specific transcription factors including MYB and their co-activators such as CBP/P300 are emerging as important targets for drug development.

Therapeutic targeting of transcription factors remains challenging due to the absence of identifiable enzymatic activities and limited knowledge regarding functionally important protein–protein interaction interfaces amenable to pharmacologic perturbation. Recent efforts have begun to develop pharmacologic approaches for their blockade, including chetomin and napthol derivatives[40–42]. In addition, proof-of-concept small-molecule inhibitors of bromo and acetyltransferase domains of CBP/P300 have been developed[18,43,44]. Finally, advances in cell transduction technology and structural biology of protein complexes have been used to design cell-penetrant peptidomimetic molecules to interfere with functionally important protein-protein interactions, including their therapeutic targeting in cancer[31,45,46].

Here, we introduce an alternative approach to interfere with the activity of transcription factors and their aberrant co-activation in cancer by disrupting the interaction of the trans-activation domain of MYB with the KIX domain of its coactivator CBP/P300. Molecular mimicry of helical domains by D-amino acid-containing retro-inverso peptides and their fusion to cationic peptides have been used to confer protease resistance and membrane penetration, respectively[25]. We found that our prototypic inhibitor MYBMIM achieves comparable binding affinity to the native MYB:CBP/P300, and directly binds to the KIX domain of CBP in vitro and in AML cells. This leads to the disassembly of the cellular MYB:CBP/P300 complex, associated with the elimination of MYB complexes from enhancers and promoters, and downregulation of MYB-dependent gene expression in AML cells. The observed activity of MYBMIM in cells can be rationalized by its accumulation in leukemia cell nuclei, where it can compete with otherwise relatively low (μM) affinity, cooperative protein-protein interactions. Ectopic overexpression of BCL2 partially rescues MYBMIM-induced apoptosis of AML cells, consistent with the essential function of MYB-induced transactivation of enhancers required for enhanced AML cell growth and survival. Correspondingly, transient MYBMIM treatment of primary patient-derived AML cells impedes their growth in two different preclinical models in vivo. Thus, MYB-MIM offers a pharmacologic strategy to block leukemogenic transcriptional coactivation as a therapy for AML and other human cancers with aberrant MYB or CBP/P300 activities.

While CBP and P300 are nearly identical in structure, they have distinct and non-redundant functions[47]. Indeed, recent study of CBP and P300 in Nup98-Hoxd13-induced leukemogenesis found that loss of p300, but not Cbp, contributes to

leukemogenesis[48]. Conversely, Cbp and p300 were cooperatively required for leukemogenesis induced by Nup98-Hoxa9 and Moz-Tif2 oncogenes[12]. Importantly, at least for AML1-ETO-induced leukemias, its leukemogenicity is in part dependent on the acetylation of AML1-ETO by CBP/P300[49]. In addition, loss-of-function mutations of CBP are present in a variety of human cancers, and recent work found a functional requirement for P300 in these CBP-deficient tumors[50]. Insofar as MYBMIM may affect the activities of the KIX domains of both CBP and P300, it is possible that MYBMIM and its drug-like derivatives may be of therapeutic utility in CBP-deficient cancers.

The KIX domain of CBP/P300 recognizes a variety of protein partners, including MYB, CREB, JUN, and MLL1, which bind to it with varying affinities and partially overlapping interaction surfaces, presumably leading to dynamically regulated and partially competitive transcription factor assemblies[51]. Given the shared physical properties of the interaction of transactivation domains of various transcription factors with the KIX domains of TAF9, MED15, and CBP/P300[52], we anticipate that similar design strategies used for MYBMIM will be useful for the modulation of their assembly for biological and therapeutic purposes. Even though binding affinity of MYB:CBP/P300 in a purified reconstituted interaction in vitro is on the μM scale, its observed effects in cells are presumably due to the TAT-directed nuclear accumulation of MYBMIM, where its extended residence time is expected to achieve specific competition of the endogenous MYB:CBP/P300 complexes. While MYBMIM exhibits specific effects on the binding and activity of MYB:CBP/P300 complex in AML cells, it is also possible that its effects may affect MLL1 and CREB interactions with CBP/P300[53,54]. Thus, MYBMIM offers a probe for the study of CBP/P300 KIX domain function and its therapeutic targeting in cancer.

We found that MYBMIM downregulated the MYB-bound BCL2 enhancer, leading to downregulation of BCL2 expression and apoptosis of leukemia cells. Insofar as this effect can be partially rescued by ectopic BCL2 overexpression, this indicates that MYB-induced dysregulation of BCL2 expression is required for MYBMIM-induced anti-leukemia effects. It is likely that altered expression of additional genes, dysregulated by leukemogenic activities of MYB, such as GFI1 for example[11,55], may also contribute to the apparent anti-leukemic efficacy of MYB-MIM. In addition, we observed that MYBMIM treatment affected enhancers and promoters enriched not only for MYB binding sites, but also for several other transcription factors, including ERG, SPI1/PU.1, CEBPA, and RUNX1 (Fig. 2). Insofar as at least some of these transcription factors can co-assemble at specific gene loci and can themselves be acetylated by CBP/P300[15,49], our findings indicate that leukemogenic transcriptional co-activation in AML may be directly related to the aberrant assembly and

**Fig. 5** MYBMIM exhibits anti-leukemia efficacy in vivo. **a** Activity of burst forming units-erythroid (BFU-E, red) and colony forming units-granulocyte/monocyte (CFU-GM, black) of CD34+ human umbilical cord progenitor cells treated with control PBS, or 20 μM TG3 or MYBMIM for 14 days. Error bars represent the standard deviation of 3 biologic replicates. **b** Representative phase photographs of CFU-GM colonies treated as indicated. **c–f** Peripheral blood count analysis of C57BL/6 J mice treated for 7 days with MYBMIM (25 mg/kg IP daily), as compared to control PBS. Bars indicate the mean and standard deviation of individual mice. **g** Plasma concentration of BIO-MYBMIM after one-time IP injection of 25 mg/kg in C57BL/6 J mice. Plasma was collected at 30 min, 2 h, 4 h, and 24 h post-injection and the concentration of BIO-MYBMIM was determined by spectrophotometric avidin assay. **h** Representative fluorescent micrographs of human-specific CD45 staining (green) and DAPI staining (blue) in femur sections of NSG mice engrafted with primary patient-derived MLL-rearranged leukemia cells and treated with MYBMIM (25 mg/kg IP daily) as compared to control PBS for 21 days upon development of peripheral leukemia, quantified in (**i**). Error bars represent standard deviation of 6 individual mice. *$p = 1.2e{-}4$, log-transformed t-test. **j** Images of fluorescent micrographs of human BCL2 (green) and DAPI (blue), quantified in (**k**). Error bars represent standard deviation of 6 individual mice. *$p = 0.3$, log-transformed t-test. **l** Kaplan–Meier survival analysis of NSG mice engrafted with primary patient-derived MLL-rearranged leukemia cells and treated 3 days post transplantation with MYBMIM (red, 25 mg/kg IP twice daily) as compared to control PBS (black) for 14 days. $n = 15$ mice per group. $p = 0.0038$, log-rank test. **m** Kaplan–Meier survival analysis of NSG mice serially transplanted with bone marrow collected from moribund mice engrafted with primary patient-derived MLL-rearranged leukemia cells treated with MYBMIM for 14 days. $n = 10$ mice per group. $p < 0.0001$, log-rank test

composition of enhanceosomes at specific gene loci. Their definition is anticipated to yield specific molecular dependencies for therapeutic modulation of aberrant transcriptional co-activation in cancer.

## Methods

**Reagents**. All reagents were obtained from Thermo Fisher unless otherwise specified. Synthetic peptides were produced by solid phase synthesis, purified by liquid chromatography, and confirmed by mass spectrometry (Tufts University Core Facility). Synthetic oligonucleotides were obtained from Eurofins. Peptides were dissolved in phosphate buffered saline at a concentration of 1 mM, as measured using optical absorbance measurements at 280 nm and extinction coefficient 1490 $M^{-1}$ $cm^{-1}$.

**Plasmids**. Bacterial expression pGEX-KIX vector encoding the KIX domain of CBP was a kind gift of Shunsuke Ishii[56]. MSCV-IRES-GFP retroviral vector encoding human *BCL2* was a gift from Takaomi Sanda[57].

**Cell culture**. The human AML lines MV-411, MOLM-13, ML-2, and HL-60 were obtained from the American Type Culture Collection (ATCC, Manassas, Virginia, USA). Umbilical cord blood was obtained from the New York Blood Center. The identity of all cell lines was verified by STR analysis (Genetica DNA Laboratories, Burlington, NC, USA) and absence of Mycoplasma sp. contamination was determined using Lonza MycoAlert (Lonza Walkersville, Inc., Walkersville, MD, USA). Cell lines were cultured in 5% $CO_2$ in a humidified atmosphere at 37 °C in RPMI medium supplemented with 10% fetal bovine serum (FBS) and antibiotics (100 U/ml penicillin and 100 µg/ml streptomycin).

**Molecular dynamics simulations**. The solution NMR structure of KIX domain of CBP bound to the transactivation domain of C-MYB (PDB code 1SB0) was used as a starting point for simulations of both L- and D-amino acid MYB-CBP complexes[23]. Specifically, the NMR structure with the lowest root-mean-square-deviation (RMSD) from the average of the ensemble of 20 solution NMR structures was selected (model 5). D-amino acid MYB peptide was built with Simulaid program using the NMR structure of protein–peptide complex and converting C-MYB peptide from L-amino acids to D-amino acids in the presence of CBP[58]. Simulations were performed using the Desmond molecular dynamics program[59]. The starting structures were solvated with 6615 and 6714 SPC water molecules, respectively, with a 5 Å buffer of water in a rectangular box. Three chloride ions were added to both systems to maintain basic electric neutrality. The OPLS3 force field was used to describe both L- and D-amino acid peptide–protein complexes[60]. For each system, a relaxation phase, with a combination of Brownian dynamics and restrained molecular dynamics phases was performed to equilibrate the systems. Periodic boundary conditions with a cutoff of 0.9 nm for both particle-mesh Ewald and Lennard–Jones interactions were used[61,62]. Each equilibrated system was then subjected to 60 ns simulations with identical parameters. Simulations were performed using the constant pressure and constant temperature (NPT) ensemble with a Berendsen thermostat and barostat. The equations of motion were integrated using RESPA with a time step of 2.0 fs for bonded and short-range nonbonded interactions, and 6.0 fs for long-range electrostatic interactions[63]. System coordinates were saved every 5 ps.

**Expression and purification of recombinant CBP KIX domain**. BL21(DE3) cells (Invitrogen) transformed with pGEX-KIX plasmid were induced at 37 °C with isopropyl β-D-1-thiogalactopyranoside for 3 h. Cells were lysed in 50 mM Tris–HCl pH 7.3, 150 mM NaCl, 0.1% Tween-20, 1 mM DTT, 5 mM EDTA, supplemented with protease inhibitors described above and sonicated for ten minutes (15 s on, 15 s off, 40% amplitude) using the Misonix probe sonicator (Qsonica, Newtown, CT). Lysate was cleared by centrifugation for 1 h at 21,800 × g at 4 °C. Cleared lysate was incubated with 4 ml glutathione agarose resin slurry (GoldBio) for 1 h at 4 °C to capture GST-KIX. Resin was washed four times with 50 mM Tris–HCl pH 7.4, 150 mM NaCl. KIX domain was cleaved from GST by incubation of resin-bound GST-KIX with 160 U thrombin (GE Healthcare) overnight at room temperature. Resin was centrifuged at 500 × g for 5 min. Supernatant containing cleaved KIX was collected and dialyzed at 4 °C against 50 mM MOPS pH 6.5, 50 mM NaCl, 10% glycerol, 1 µM tris-2-carboxyethylphosphine. Cleaved KIX was purified using a linear gradient of 50 mM to 1 M NaCl by cation exchange chromatography using MonoS 5/50 GL column (GE Healthcare). Fractions containing purified KIX were dialyzed against 50 mM potassium phosphate pH 5.5, 150 mM NaCl, 10 µM tris-2-carboxyethylphosphine, 30% glycerol, and stored at −80 °C.

**Microscale thermophoresis (MST)**. Binding of purified recombinant KIX with FITC-conjugated peptides was measured using Monolith NT.115 (NanoTemper Technologies). Assays were conducted in 50 mM sodium phosphate, 150 mM NaCl, 0.01% NP-40, pH 5.5. FITC-conjugated peptides (FITC-MYB at 250 nM, FITC-MYBMIM at 500 nM, FITC-TAT at 500 nM, FITC-TG1 at 500 nm, FITC-TG2 at 500 nm, and FITC-TG3 at 500 nm) were mixed with 16 increasing

concentrations of KIX (0.0015–50 µM, 1:1 serial dilutions) and loaded into MST Premium Coated capillaries. MST measurements were recorded at room temperature for 10 s per capillary using fixed IR-laser power of 80% and LED excitation power of 40–50%.

**Confocal microscopy**. Confocal imaging was performed using the Zeiss LSM880 confocal microscope and 40X objective with 1.5 µm z-stack images. Cells were applied to a poly-L-lysine-coated chambered Nunc Lab-tek II coverslip and incubated for 2 h at 37 °C. FITC-conjugated MYBMIM was added to cell suspensions at a concentration of 50 nM and incubated for 1 h at 37 °C. Cells were counter-stained using Hoechst 33342 and Mitotracker Red CMX ROS (MProbes) for 10 min at a final dilution of 1:10,000 prior to imaging.

**Western blot analysis**. Cells were lysed in RIPA buffer (Thermo Fisher) supplemented with a protease inhibitor mix comprised of AEBSF (0.5 mM concentration, Santa Cruz, SC-202041B), Bestatin (0.01 mM, Fisher/Alfa Aesar, J61106-MD), Leupeptin (0.1 mM, Santa Cruz, SC-295358B), and Pepstatin (0.001 mM, Santa Cruz, SC-45036A). Lysates were mechanically disrupted using Covaris S220 adaptive focused sonicator, according to the manufacturer's instructions (Covaris, Woburn, CA). Lysates were cleared by centrifugation for 15 min at 18,000 × g and clarified lysates were quantified using the bicinchoninic acid assay (Pierce). Clarified lysates (20 µg of protein) were resolved using sodium dodecyl sulfate-polyacrylamide gel electrophoresis, and electroeluted using the Immobilon FL PVDF membranes (Millipore, Billerica, MA, USA). Membranes were blocked using the Odyssey Blocking buffer (Li-Cor, Lincoln, Nebraska, USA). The following primary antibodies were used as indicated: anti-MYB (1:1000, 05-175, Millipore), anti-CBP (1:1000, PA1-847, Invitrogen), anti-BCL2 (1:1000, 200-401-222, Rockland), anti-β actin (1:1000, 8H10D10, Cell Signaling). Anti-CBP antibody is known to cross-react with P300[64]. Blotted membranes were visualized using secondary antibodies conjugated to IRDye 800CW or IRDye 680RD (Goat anti-rabbit, 1:15,000, and goat anti-mouse, 1:15,000) and the Odyssey CLx fluorescence scanner, according to manufacturer's instructions (Li-Cor, Lincoln, Nebraska, USA).

**Co-immunoprecipitation analysis**. In total 7.5 µg of anti-MYB antibodies (EP769Y, Abcam) were conjugated to 1 mg M-270 Epoxy-coated magnetic beads (Invitrogen) according to manufacturer's instructions. In total $2 \times 10^7$ MV-411 cells were collected and washed in cold PBS. Washed cell pellets were resuspended in 2 ml cold lysis buffer (50 mM Tris–HCl pH 7.4, 150 mM NaCl, 0.5 mM EDTA, 1 mM DTT, 0.5% Triton X-100, 10% glycerol supplemented with protease inhibitors described above) and incubated on ice for 10 min. Cells centrifuged for 5 min at 2000 × g at 4 °C. Supernatant was clarified by centrifugation for 15 min at 18,000 × g at 4 °C. Cleared lysate was added to 1 mg beads, and MYBMIM was added to a final concentration of 20 µM. Immunoprecipitation proceeded for 3 h at 4 °C with rotation. Beads were washed with 1 ml cold lysis buffer twice. Proteins were eluted in 30 µl EB buffer (Invitrogen) for 5 min at room temperature with agitation, and eluate was neutralized with 2 µl 1 M Tris pH 11. Samples were prepared for western blot by addition of Laemmli buffer with 50 mM DTT and incubation at 95 °C for 5 min. The presence of MYB and CBP/P300 was identified by western blot as described.

**Streptavidin affinity purification**. Streptavidin magnetic beads (Pierce) were washed with PBS with 0.5% BSA twice prior to use. Biotinylated MYBMIM (BIO-MYBMIM) was conjugated to 100 µl streptavidin bead slurry (1.5 mg beads, binding capacity 3500 pmol biotinylated fluorescein per mg) by incubation at room temperature for 1 h in 1 ml PBS with 0.5%. Peptide-conjugated beads were washed twice in 1 ml PBS with 0.5% BSA. $1 \times 10^7$ cells were collected and washed in cold PBS. Washed cell pellets were resuspended in 1 ml of cold lysis buffer (50 mM Tris–HCl pH 7.4, 150 mM NaCl, 0.5 mM EDTA, 1 mM DTT, 0.5% Triton X-100, 10% glycerol supplemented with protease inhibitors described above) and incubated on ice for 10 min. Cells were centrifuged for 5 min at 2000 × g at 4 °C. Supernatant was clarified by centrifugation for 15 min at 18,000 × g at 4 °C. PBS with 0.5% BSA was removed from peptide-conjugated streptavidin bead slurry, lysate was added to 1 mg beads, and affinity purification proceeded for 3 h at 4 °C. For peptide competition, MYBMIM or RI-TAT was added at 20-fold molar excess at the time of affinity purification. Beads were washed twice with 1 ml cold lysis buffer. Bound proteins were eluted by adding 40 µl Laemmli buffer with 50 mM DTT and incubated for 5 min at 95° C. The presence of CBP/P300 was identified by Western blot as described.

**Chromatin immunoprecipitation and sequencing (ChIP-seq)**. ChIP was performed as previously described[65]. Briefly, cells were fixed in 1% formalin in phosphate-buffered saline (PBS) for 10 minutes at room temperature. Glycine (125 mM final concentration) and Tris-HCl pH 8 (100 mM final concentration) were added to the cells and cells were washed twice in ice-cold PBS and resuspended in sodium dodecyl sulfate (SDS) lysis buffer (1% SDS, 10 mM EDTA, 50 mM Tris–HCl, pH 8.1). Lysates were sonicated using the Covaris S220 adaptive focused sonicator to obtain 100–500 bp chromatin fragments (Covaris, Woburn, CA). Lysates containing sheared chromatin fragments were resuspended in 0.01% SDS,

1.1% Triton-X100, 1.2 mM EDTA, 16.7 mM Tris–HCl, pH 8.1,167 mM NaCl. Lysates and antibody-coupled beads were incubated over night at 4 °C. Precipitates were washed sequentially with Mixed Micelle Wash Buffer (15 ml 5 M NaCl −150 mM Final, 10 ml 1 M Tris–Cl pH 8.1, 5 ml 0.5 M EDTA, pH 8.0, 40 ml 65% w/v sucrose, 1 ml 10% NaN₃, 25 ml 20% Triton X-100, 10 ml 10% SDS, Add dH₂O to 500 ml), LiCl washing solution (0.5% deoxycholic acid, 1 mM EDTA, 250 mM LiCl, 0.5% NP-40, 10 mM Tris–Cl pH 8.0, 0.2% NaN3) and then TBS buffer (20 mM Tris–Cl pH 7.4, 150 mM NaCl). Elution performed in elution buffer (1% SDS, 0.1 M NaHCO3). ChIP-seq libraries were generated using the NEBNext ChIP-seq library prep kit following the manufacturer's protocol (New England Biolabs, Ipswich, MA, USA). Libraries were sequenced on the Illumina HiSeq 2500 instruments, with 30 million 2 × 50 bp paired reads.

For ChIP-seq analysis, reads were quality and adapter trimmed using 'trim_galore' before aligning to human genome assembly hg19 with bwa mem using the default parameters. Aligned reads with the same start position and orientation were collapsed to a single read before subsequent analysis. Density profiles were created by extending each read to the average library fragment size and then computing density using the BEDTools suite. Enriched regions were discovered using MACS 2.0 and scored against matched input libraries. Genomic 'blacklisted' regions were filtered (http://www.broadinstitute.org/~anshul/projects/encode/rawdata/blacklists/hg19-blacklist-README.pdf) and remaining peaks within 1 kb were merged. Read density normalized by sequencing depth was then calculated for the union of peaks, and the MYBMIM and control samples were compared using Welch's t-test.

**Chromatin immunoprecipitation and quantitative PCR (ChIP-PCR).** For H3K27Ac ChIP-PCR, MV-411 cells were treated with 20uM MYBMIM or TG3 for 12 h and then cross-linked with 1% formaldehyde for 10 min at room temperature. Cross-linking was ended by the addition of 1/20 volume of 2.5 M Glycine for 5 min at room temperature followed by cell lysis and sonication (E220 Covaris sonicator) to obtain 100- to 500-bp chromatin fragments. H3K27Ac Rabbit polyclonal antibody (Abcam, #4729) was conjugated to Protein A and G Dynabeads per manufacturer's instructions (Thermo Fischer Scientific). Lysates were incubated overnight at 4 °C with antibody-conjugated beads in suspension. Precipitates were then washed sequentially with cold washing solution (1% NP-40, 1 mm EDTA, 50 mM Hepes-KOH, pH 7.6, 500 mM LiCl, 0.7% Na-Deoxycholate) and then washing solution (50 mM Tris–HCL, pH 8.0, 10 mM EDTA, 50 mM NaCl), then eluted in elution buffer (50 mM Tris–HCL, pH 8.0, 10 mM EDTA, 1% SDS). Reversal of crosslinks in elution buffer overnight at 65 °C followed by digestion of RNA and protein using RNase (Roche, Catalog No. 111119915-001) and Proteinase K (Roche, Catalog No. 03115828001). DNA purification was performed using PureLink PCR Purification Kit per manufacturer's protocol (Invitrogen). qPCR was performed as described below.

**RNA sequencing (RNA-seq).** Reads were quality and adapter trimmed using "trim_galore" before aligning to human assembly hg19 with STAR v2.5 using the default parameters. Coverage and post-alignment quality were assessed using the Picard tool CollectRNASeqMetrics (http://broadinstitute.github.io/picard/). Read count tables were created using HTSeq v0.6.1. Normalization and expression dynamics were evaluated with DESeq2 using the default parameters.

**Cell viability analysis.** Cells were resuspended and plated at a concentration of 2 × 10⁵ cells in 200 μl in 96-well tissue culture plates. Media with peptides was replaced every 48 h. To assess the number of viable cells, cells were resuspended in PBS and 10 μl of suspension was mixed in a 1:1 ratio with 0.4% Trypan Blue (Thermo Fisher) and counted using a hemacytometer (Hausser Scientific, Horsham, PA, USA). To assess viability using an ATP-based assay, cell viability was assessed using the CellTiter-Glo Luminescent Viability assay, according to the manufacturer's instructions (Promega). Luminescence was recorded using the Infinite M1000Pro plate reader using integration time of 250 ms (Tecan).

**Flow cytometric analysis of apoptosis.** Cells were resuspended to a concentration of 1 × 10⁶ cells were plated in triplicate in a 12-well tissue culture plate. For assessment of annexin V staining, cells were washed with PBS and then resuspended in PBS with Annexin V-APC (BioLegend) and propidium iodide at a dilution of 1:1000. For intracellular detection of cleaved caspase 3, cells were fixed and permeabilized using the BD Cytofix/Cytoperm Fixation/Permeabilization solution according to the manufacturer's instructions (BD Biosciences). Cells were then stained using the Alexa Fluor 647-conjugated anti-active caspase-3 (BD Biosciences) at a dilution of 1:50. Cells were incubated for 30 minutes room temperature in the dark, washed, and then analyzed using the BD LSRFortessa cell analyzer. For assessment of differentiation, cells were stained using the anti-human CD14 PE at a dilution of 1:20 (Affymetrix eBiosciences) and anti-human CD66b at a dilution of 1:20 (Affymetrix eBiosciences).

**Giemsa staining of cells for morphology.** Cells were resuspended to a concentration of 1 × 10⁶ cells in 1 milliliter of PBS. Using the benchtop Cytospin Centrifuge instrument (ThermoFisher Scientific), 200 μl of the cell suspension was applied in white clipped Cytofunnels (ThermoFisher Scientific) to glass microscope slides (2 × 10⁵ cells/slide). Dip Quick Stain (J-322, Jorgensen Laboratories, Inc) was used for per manufacturer's protocol for the polychromic stain of cells.

**Quantitative RT-PCR.** RNA was isolated using Trizol reagent according to the manufacturer's instructions (Life Technologies). Complementary DNA was synthesized using the SuperScript III First-Strand Synthesis system according to the manufacturer's instructions (Invitrogen). Quantitative real-time PCR was performed using the KAPA SYBR FAST PCR polymerase with 20 ng template and 200 nM primers, according to the manufacturer's instructions (Kapa Biosystems, Wilmington, MA, USA). PCR primers are listed in Supplementary Table 4. Ct values were calculated using ROX normalization using the ViiA 7 software (Applied Biosystems).

**Retrovirus production and cell transduction.** The MIG-BCL2 vector was packaged using pUMVc and pCMV-VSVG vectors in HEK 293T cells and the FuGENE 6 transfection reagent, according to manufacturer's instructions (Promega). Virus supernatant was collected at 48 and 72 h post transfection, pooled, filtered and stored at −80 °C. Cells were transduced with virus particles at a multiplicity of infection of 1 by spin inoculation for 90 min at 3500 rpm at 35 °C in the presence of 8 μg/ml hexadimethrine bromide. Two days after transduction, cells were isolated using fluorescence-activated cell sorting (FACSAria III, BD Bioscience, San Jose, CA, USA).

**Blood progenitor colony forming assays.** Mononuclear cells were isolated from cord blood using Ficoll-Paque PLUS density centrifugation and enriched for CD34⁺ cells using the CD34 MicroBead Kit UltraPure, according to the manufacturer's instructions (Miltenyi Biotech). CD34⁺ cells were resuspended to a concentration of 1 × 10⁵ cells/ml. Methocult H4034 Optimum (Stemcell Technologies, Catalog no. 04034 with FBS, BSA and recombinant cytokines rhSCF, rhGM-CSF, rhG-CSF, rhIL3, and rhErythropoietin) semi-solid media was used for the growth of hematopoietic progenitor cells in colony-forming units. Methocult and CD34 + cells were mixed in a ratio of 1:10 (cells:Methocult) for a final cell concentration and plated at 1000 cells/dish. TG3 or MYBMIM peptides were added to this solution for a final concentration of 20 μM. Mixture was vortexed for 30 s and incubated at room temperature for 5 min. Using a blunt end 18 G needle, 1.1 ml of the solution was added to a 35 × 10 mm² dish and then tilted to cover. Peptide treatment conditions were plated in biological triplicates. 35 × 10 mm dishes were placed into a larger 100 × 15 mm² dish with one 35 × 10 mm² dish filled with sterile water. Dishes were incubated at 37 °C with 5% CO2 for 14 days. Both erythroid progenitor and granulocyte-macrophage progenitors were observed and quantified. Brightfield microscopy and CFU-Gm and BFU-E colony images were obtained using ×10 and ×20 magnification using the Zeiss inverted microscope.

**Mouse studies.** All mouse experiments were carried out in accordance with the Memorial Sloan Kettering Institutional Animal Care and Use Committee. For toxicity studies, female C57BL/6J mice (The Jackson Laboratory, Bar Harbor, Maine, USA) were treated with MYBMIM peptide suspended in PBS and administered daily through intraperitoneal injection at a dose of 25 mg/kg for a total of 7 days. Mice were sacrificed at the end of treatment for hematologic, biochemical and histologic analyses. For pharmacokinetic studies, C57BL/6J mice (The Jackson Laboratory, Bar Harbor, Maine, USA) were treated with a single IP injection of 25 mg/kg BIO-MYBMIM, and blood plasma was collected 30 min, 2 h, 4 h, and 24 h post-injection. Quantification of BIO-MYBMIM was measured using the Quant-Tag Biotin kit (Vector Labs, cat. # BDK-2000) following the manufacturer instructions. For patient-derived xenografts, two hundred thousand primary AML MLL-rearranged leukemia cells were suspended in 200 ml of PBS and transplanted via tail vein injection into 8-week-old sublethally irradiated (200 rad) female NOD.Cg-Prkdc(scid)Il2rg(tm1Wjl)/SzJ mice (The Jackson Laboratory, Bar Harbor, Maine, USA). Recipient mice were maintained on antibiotic supplementation in chow (0.025% trimethoprim, 0.124% sulfamethoxazole, Sulfatrim). Three days after transplant, mice were randomly assigned to experimental treatment groups. MYBMIM peptide suspended in PBS was administered twice daily through intraperitoneal injection at a dose of 25 mg/kg per injection. Mice were treated from days 3–17 of this study for a total of 14 days and then monitored daily with clinical examination for survival analysis.

**Immunofluorescence staining.** The immunofluorescence detection was performed with a Discovery XT system (Ventana Medical Systems). Tissue sections were blocked first for 30 min in Mouse IgG Blocking reagent (Vector Labs; cat. # MKB-2213) in PBS. The primary antibody incubation was performed with either mouse monoclonal Anti Human CD45 (Dako, Catalog No. M0701, 2.5 μg/ml) or rabbit polyclonal Anti BCL2 (Ventana, Catalog No. 790-4604, 0.24 μg/ml) for 6 h followed by 60 minutes incubation with a biotinylated mouse secondary antibody (Vector Labs, MOM Kit BMK-2202), at 5.75 μg/ml (1:200 dilution). The detection was performed with Secondary Antibody Blocker, Blocker D, Streptavidin-HRP D (Ventana Medical Systems), followed by incubation with Tyramide-Alexa Fluor 488 (Invitrogen, cat. #T20922).

**Data availability**. The data discussed in this publication have been deposited in NCBI's Gene Expression Omnibus and are accessible through GEO Series accession numbers GSE94242 and GSE107078. Supplementary data files are additionally available at Zenodo (https://doi.org/10.5281/zenodo.1098327).

**Statistical Analysis**. For comparisons between two sample sets, statistical analysis of means was performed using two-tailed, unpaired Student's *t*-tests. Survival analysis was done using the Kaplan–Meier method, as assessed using a log-rank test. For gene expression analysis, statistical significance was assessed using paired Student's *t*-tests.

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

## Acknowledgements

We thank Alejandro Gutierrez, Leo Wang, and Marc Mansour for helpful discussions, and Antoine Gruet and Yang Li for technical assistance. This work was supported by the NIH R21 CA188881, R01 CA204396, DK101989, CA193842, CA176745, CA066996, K08 CA191091, P30 CA008748, T32 GM073546, Burroughs Wellcome Fund, Josie Robertson Investigator Program, American Society of Hematology, Rita Allen Foundation, Alex's Lemonade Stand Foundation, Gabrielle's Angel Foundation, Leukemia & Lymphoma Society, Bill and Melinda Gates Foundation, and Mr. William H. and Mrs. Alice Goodwin and the Commonwealth Foundation for Cancer Research and the Center for Experimental Therapeutics at MSKCC. K.R. was supported by the Charles Trobman Scholarship. A.K. is the Damon Runyon-Richard Lumsden Foundation Clinical Investigator.

## Author contributions

K.R. performed experiments, analyzed data and designed study; L.F., F.B, T.G., G.M., M. K., A.K., S.A., E.S., E.deS., B.K., R.K. performed experiments and analyzed data; A.K. analyzed data and designed study. K.R. and A.K. wrote the manuscript with contributions from other co-authors.

## Additional information

**Competing interests:** The authors declare no competing financial interests.

