## [Peer Review File · Nature Communications]

Reviewers' comments:

Reviewer #1 (Remarks to the Author):

In this manuscript by Ramaswamy and colleagues, the authors design and test a novel D-amino acid-containing retro-inverso peptide mimetic to interfere with the MYB:CBP/P300 complex. This complex has been shown by other groups to be integral to the survival of AML cells both in vivo and in vitro, but efforts to target this for therapeutic potential have thus far been challenging. Given CBP inhibitors are likely to be toxic, specifically interfering with the MYB-CBP interaction is hypothetically a much more targeted approach, with a potential therapeutic window. The rationale for targeting the MYB E308 is highly innovative, given the Booreana mouse strain that has a mutation of this amino acid shows impaired MYB:KIX domain interactions. The paper is clear and well-written, and on the whole, the data supports the underlying hypothesis. Conceptually this is a novel and inventive approach, and I expect this study will have an important place in the field of peptidomimetics and those interested in MYB and MYB-driven malignancies such as acute leukemias. I have a number of specific questions, and some suggestions that could improve the paper:

Line 107, figure 1C. '...MYBMIM bound to the CBP-KIX domain in a MYB peptide-dependent manner...' The terminology is confusing as it suggests MYBMIM will only bind CBP-KIX in the presence of wild-type MYB. Is it possible to dissociate MYBMIM from CBP-KIX with MYB, and/or vice versa? Given the differences in affinities shown in fig 1C, how do the authors explain the therapeutic effect in vitro?

Line 138, fig 2A: although molecular dynamic simulations would predict the TG3 mutant cannot bind CBP, the authors do need to validate this experimentally. For instance, one would predict TG3 cannot dissociate MYB from CBP (as in fig 1f).

Line 172, figure 2f: addition of H3K27ac ChIP-seq would be compelling to assess MYB driven enhancer formation genome-wide.

Fig 3B – the lack of induction of differentiation is somewhat surprising given the data from Zuber Genes and Development paper, where a differentiation program is initiated on MYB knockdown in MLL-AF9 leukemias. This was predominantly to neutrophil lineage, so have the authors looked for CD11b induction?

Fig 3C – can the authors show the apoptotic effects of MYBMIM are due to on-target MYB-CBP inhibition?

Fig 4: the survival benefit in vivo is minimal. Some mention of why this might be the case should be addressed in the discussion. For instance, no mention is made of peptide PK in vivo. Have the authors considered assessing tissue penetration of the peptide using a tag? For instance, the FITC-tagged MYBMIM could be administered and leukemic bone marrow infiltrate assessed by FACS.

Minor points:

Fig 1 D : higher magnification images are required to identify localization.

Figure 2F: higher magnification of the BCL2 promoter peaks would make these easier to visualize.

Figure 3D legend, cell type needs specifying.

It would be courteous to quote Fawell et al PNAS 1994 in the design of the TAT peptide approach, line 96.

Reviewer #2 (Remarks to the Author):

This manuscript by Kentsis and coworkers describes the discovery and use of a peptidomimetic (MYBMIM) that functions to prevent the interaction of the trans-activation domain of MYB with the KIX domain of CBP/p300. MYB activity has been shown to be important for maintenance of acute myeloid leukemia but has been difficult to target. This clearly written manuscript provides an important advance in the targeting of MYB trans-activation activity. In so doing, the authors show that inhibition of MYB trans-activation can impede leukemia growth. The authors go on to investigate the mechanism of MYBMIM cellular activity and observe that MYB occupancy is reduced and MYB-dependent genes have lower expression, including BCL2. Importantly, the authors note in discussion that MYBMIM might not be specific inhibitor of the MYB:CBP/p300 interaction, rather it is a specific probe of CBP/P300 KIX domain function. I would have liked experimental effort to understand whether MYBMIM disrupts other CBP/P300 KIX domain interactions, but I believe it is outside the scope of the present work. Given the limited treatment options for patients with AML and the limited effectiveness of those treatments, advancing new therapeutic strategies for AML is of high interest. Furthermore, approaches to targeting transcription factor activity is of high interest. Overall, I support publication of this manuscript after addressing my specific comments below.

1. A key result in the paper is that MYBMIM targets the MYB:CBP/p300 interaction to inhibit AML cell proliferation. There is sufficient evidence that MYBMIM inhibits the MYB:CBP/p300 interaction in cells (Figs 1e,f), but the main evidence to support the connection between CBP/p300 binding and AML proliferation, SAR with TG3, could be strengthened. First, I recommend including in vitro binding data for TG3 (microscale thermophore assay) and similarly full dose-response data for TG3 and MYBMIM in MOLM-13 and/or MV4;11 (viability or pro-apoptotic activity). Second, I recommend including similar SAR data with additional single and double MYBMIM mutants. One might expect such mutants to have intermediate activity between MYBMIM and TG3. Together, this set of 4 to 5 peptidomimetics could enable a correlation analysis between in vitro CBP binding and cellular antiproliferative/proapoptotic activity to strengthen the connection between CBP/p300 binding and inhibition of AML cell proliferation.

2. Improvements could be made to the presentation of the in vivo data.

- a. I would like to understand how the 25 mpk once-daily IP and twice-daily IP doses were determined, as well as the treatment duration for each study. Were MTD and PK studies first performed to identify tolerated dosing and exposure levels that correlate with cell culture activity?
- b. I recommend placing Figures 4c-f (peripheral blood analysis) after the efficacy results. The efficacy results establish the justification for the doses selected for this analysis.
- c. For further safety assessment, please include the body weight data (% of original) for each in vivo study during the duration of treatment.
- d. Given the modest reduction in BCL2 protein levels seen, it would be useful to determine that BCL2 protein levels decrease in MOLM-13 and MV4;11 cells, in addition to mRNA levels.

3. Small edits

- a. Add key to define each bar in Supplementary Fig. 5.
- b. Line 482 – header ChIP-PCR rather than ChIP-seq
- c. Line 586 – more clarification needed on the actual dose used – 50 mpk once daily or 25 mpk twice daily as indicated in the text and figure?

We are pleased that the referees and the editor share our enthusiasm for “Peptidomimetic blockade of MYB in acute myeloid leukemia.” Below are point-by-point responses to all of the referees’ comments, which we have now incorporated into the enclosed revised manuscript, which we believe will be of high interest to the readers of *Nature Communications*.

Reviewer #1 (Remarks to the Author):

In this manuscript by Ramaswamy and colleagues, the authors design and test a novel D-amino acid-containing retro-inverso peptide mimetic to interfere with the MYB:CBP/P300 complex. This complex has been shown by other groups to be integral to the survival of AML cells both in vivo and in vitro, but efforts to target this for therapeutic potential have thus far been challenging. Given CBP inhibitors are likely to be toxic, specifically interfering with the MYB-CBP interaction is hypothetically a much more targeted approach, with a potential therapeutic window. The rationale for targeting the MYB E308 is highly innovative, given the Booreana mouse strain that has a mutation of this amino acid shows impaired MYB:KIX domain interactions. The paper is clear and well-written, and on the whole, the data supports the underlying hypothesis. Conceptually this is a novel and inventive approach, and I expect this study will have an important place in the field of peptidomimetics and those interested in MYB and MYB-driven malignancies such as acute leukemias. I have a number of specific questions, and some suggestions that could improve the paper:

- 1) *Line 107, figure 1C. ‘...MYBMIM bound to the CBP-KIX domain in a MYB peptide-dependent manner...’ The terminology is confusing as it suggests MYBMIM will only bind CBP-KIX in the presence of wild-type MYB. Is it possible to dissociate MYBMIM from CBP-KIX with MYB, and/or vice versa? Given the differences in affinities shown in fig 1C, how do the authors explain the therapeutic effect in vitro?*

Response: MYBMIM binding is dependent on the MYB, not TAT, portion of the peptide, and we have modified the text accordingly (page 6). Based on the direct binding experiments in vitro, and dissociation of MYB:CBP complex immunoprecipitated from cell extracts, we believe that the binding of MYBMIM to CBP is reversible, and the therapeutic effect can be attributed to its accumulation in leukemia cell nuclei. We have revised the text to clarify this point (page 6, 13, 14).

- 2) *Line 138, fig 2A: although molecular dynamic simulations would predict the TG3 mutant cannot bind CBP, the authors do need to validate this experimentally. For instance, one would predict TG3 cannot dissociate MYB from CBP (as in fig 1f).*

Response: We have now measured the binding affinity of TG3, as well as a newly designed version TG1, which lacks the leucine hydrophobic interaction, and TG2 lacking the leucine hydrophobic interaction and one of the terminal salt bridge interactions (peptide information detailed in Supplementary Table 1). These new results are shown in revised Figure 1c and 1f, and confirm that TG3 is a suitable inactive analogue of MYBMIM.

- 3) *Line 172, figure 2f: addition of H3K27ac ChIP-seq would be compelling to assess MYB driven enhancer formation genome-wide.*

Response: The revised manuscript includes genome-wide analysis the effects of MYBMIM in H3K27Ac, and its association with MYB occupancy (Figure 3b, text clarified on page 8). In

particular, we found significant changes in H3K27Ac occupancy at specific loci that co-localize with changes in occupancy at GFI1, in addition to the BCL2 enhancer locus already shown (Supplementary figure 4).

- 4) *Fig 3B – the lack of induction of differentiation is somewhat surprising given the data from Zuber Genes and Development paper, where a differentiation program is initiated on MYB knockdown in MLL-AF9 leukemias. This was predominantly to neutrophil lineage, so have the authors looked for CD11b induction?*

Response: We have now included the analysis of CD11b expression in response to MYBMIM with no significant change in CD11b expression as compared to controls (Supplementary Figure 5), and have revised the manuscript to clarify this point (page 9).

- 5) *Fig 3C – can the authors show the apoptotic effects of MYBMIM are due to on-target MYB-CBP inhibition?*

Response: We concluded that MYBMIM-induced apoptosis is due to disassembly of MYB:CBP complex insofar as the TG3 inactive analogue of MYBMIM does not have this effect and does not appreciably bind to CBP (revised Figures 1c and f), MYBMIM induces significant reduction of BCL2 expression, and MYBMIM-induced apoptosis can be partially rescued by ectopic BCL2 expression. We agree with the reviewer that these results do not exclude the possibility that MYBMIM can also affect additional as of yet unknown cellular factor, and have modified the text to clarify this point (revised text on page 9).

- 6) *Fig 4: the survival benefit in vivo is minimal. Some mention of why this might be the case should be addressed in the discussion. For instance, no mention is made of peptide PK in vivo. Have the authors considered assessing tissue penetration of the peptide using a tag? For instance, the FITC-tagged MYBMIM could be administered and leukemic bone marrow infiltrate assessed by FACS.*

Response: We have now measured the pharmacokinetics of biotinylated MYBMIM in mice, and include these results in the revised manuscript (revised Figure 4g), as well as a discussion of this point (revised text on page 10).

- 7) *Minor points:*

Fig 1 D: higher magnification images are required to identify localization.

Response: We have now included images with higher magnification to better highlight the nuclear localization of FITC-MYBMIM.

- 8) *Figure 2F: higher magnification of the BCL2 promoter peaks would make these easier to visualize.*

Response: The magnification of figure 2f has been adjusted to highlight this promoter peak.

- 9) *Figure 3D legend, cell type needs specifying.*

Response: The cell in figure 3d had been added (MV411 cells).

10) It would be courteous to quote Fawell et al PNAS 1994 in the design of the TAT peptide approach, line 96.

Response: This reference has been added to the revised manuscript.

Reviewer #2 (Remarks to the Author):

This manuscript by Kentsis and coworkers describes the discovery and use of a peptidomimetic (MYBMIM) that functions to prevent the interaction of the trans-activation domain of MYB with the KIX domain of CBP/p300. MYB activity has been shown to be important for maintenance of acute myeloid leukemia but has been difficult to target. This clearly written manuscript provides an important advance in the targeting of MYB trans-activation activity. In so doing, the authors show that inhibition of MYB trans-activation can impede leukemia growth. The authors go on to investigate the mechanism of MYBMIM cellular activity and observe that MYB occupancy is reduced and MYB-dependent genes have lower expression, including BCL2. Importantly, the authors note in discussion that MYBMIM might not be specific inhibitor of the MYB:CBP/p300 interaction, rather it is a specific probe of CBP/P300 KIX domain function. I would have liked experimental effort to understand whether MYBMIM disrupts other CBP/P300 KIX domain interactions, but I believe it is outside the scope of the present work. Given the limited treatment options for patients with AML and the limited effectiveness of those treatments, advancing new therapeutic strategies for AML is of high interest. Furthermore, approaches to targeting transcription factor activity is of high interest. Overall, I support publication of this manuscript after addressing my specific comments below.

1. A key result in the paper is that MYBMIM targets the MYB:CBP/p300 interaction to inhibit AML cell proliferation. There is sufficient evidence that MYBMIM inhibits the MYB:CBP/p300 interaction in cells (Figs 1e,f), but the main evidence to support the connection between CBP/p300 binding and AML proliferation, SAR with TG3, could be strengthened. First, I recommend including in vitro binding data for TG3 (microscale thermophore assay) and similarly full dose-response data for TG3 and MYBMIM in MOLM-13 and/or MV4;11 (viability or pro-apoptotic activity). Second, I recommend including similar SAR data with additional single and double MYBMIM mutants. One might expect such mutants to have intermediate activity between MYBMIM and TG3. Together, this set of 4 to 5 peptidomimetics could enable a correlation analysis between in vitro CBP binding and cellular antiproliferative/proapoptotic activity to strengthen the connection between CBP/p300 binding and inhibition of AML cell proliferation.

Response: We have now designed additional versions of MYBMIM: TG1, lacking the leucine hydrophobic interaction, and TG2 lacking the leucine hydrophobic interaction and one of the terminal salt bridge interactions (peptide information included in Supplementary Table 1). We assessed these analogues, along with TG3 and MYBMIM, using direct binding assays in vitro, and phenotypic assays in cells. These new experimental results are presented in revised Figures 1c and 3a, and confirm the proposed mechanism of MYBMIM-induced disassembly of the cellular MYB:CBP complex and downregulation of leukemogenic gene expression.

2. Improvements could be made to the presentation of the in vivo data.

a. I would like to understand how the 25 mpk once-daily IP and twice-daily IP doses were determined, as well as the treatment duration for each study. Were MTD and PK studies first

performed to identify tolerated dosing and exposure levels that correlate with cell culture activity?

Response: We have now measured the pharmacokinetics of biotinylated MYBMIM in mice, and include these results in the revised manuscript (revised Figure 4g).

b. I recommend placing Figures 4c-f (peripheral blood analysis) after the efficacy results. The efficacy results establish the justification for the doses selected for this analysis.

Response: After adding the PK data with the Biotinylated-MYBMIM, we have revised Figure 4 to include this new data.

c. For further safety assessment, please include the body weight data (% of original) for each in vivo study during the duration of treatment.

Response: We have included this data in the revised manuscript (Supplementary Table 5).

d. Given the modest reduction in BCL2 protein levels seen, it would be useful to determine that BCL2 protein levels decrease in MOLM-13 and MV4;11 cells, in addition to mRNA levels.

Response: We have now assessed BCL2 protein abundance upon treatment of cells with MYBMIM (Supplementary Figure 9).

3. Small edits

a. Add key to define each bar in Supplementary Fig. 5.

Response: The key has been added to Supplementary Figure 5.

b. Line 482 – header ChIP-PCR rather than ChIP-seq

Response: This header has been adjusted to indicate ChIP-PCR.

c. Line 586 – more clarification needed on the actual dose used – 50 mpk once daily or 25 mpk twice daily as indicated in the text and figure?

Response: The dosing (25mg/kg IP twice daily) has been detailed in the revised text and figure.

REVIEWERS' COMMENTS:

Reviewer #1 (Remarks to the Author):

I am satisfied with the point-by-point responses to my queries.

Reviewer #2 (Remarks to the Author):

The authors have adequately addressed my questions and suggestions and I support publication.

Minor edits:

1. Please add concentrations used for TIG3 and MYBMIM to Figure 1f legend. I assume the concentrations were 20 micromolar for consistency with other experimental results?
2. Please change wording of text for lines 205/206. The decrease in BCL2 protein levels is modest at best, especially for MOLM13.